# EGCG Restricts PRRSV Proliferation by Disturbing Lipid Metabolism

Peng-Wei Yu,[a,c,d] Peng-Fei Fu,[a,c,d] Lei Zeng,[a,c,d] Yan-Li Qi,[a,c,d] Xiu-Qing Li,[a,c,d] Qi Wang,[a,c,d] Guo-Yu Yang,[c,d] Hua-Wei Li,[b] Jiang Wang,[a,c,d] (ID) Bei-Bei Chu,[a,c,d,e] (ID) Meng-Di Wang[b]

aCollege of Veterinary Medicine, Henan Agricultural University, Zhengzhou, Henan Province, People's Republic of China
bSchool of Food and Bioengineering, Henan University of Animal Husbandry and Economy, Zhengzhou, Henan Province, People's Republic of China
cKey Laboratory of Animal Biochemistry and Nutrition, Ministry of Agriculture and Rural Affairs of the People's Republic of China, Zhengzhou, Henan Province, People's Republic of China
dKey Laboratory of Animal Growth and Development, Zhengzhou, Henan Province, People's Republic of China
eInternational Joint Research Center of National Animal Immunology, Henan Agricultural University, Zhengzhou, Henan Province, People's Republic of China

Peng-Wei Yu and Peng-Fei Fu contributed equally to this article. Author order was determined by the corresponding author after negotiation.

**ABSTRACT** Porcine reproductive and respiratory syndrome virus (PRRSV) infection leads to late-term reproductive failure and respiratory illness that affect the global swine industry. Epigallocatechin gallate (EGCG) is a polyphenolic compound from green tea that exerts antiviral activity against diverse viruses. This study aimed to report an uncharacterized mechanism of how EGCG restricted PRRSV proliferation. EGCG showed no significant effects on cell viability, cell cycle progression, and apoptosis in porcine alveolar macrophages and MARC-145 cells. The treatment of cells with EGCG attenuated the replication of both highly pathogenic and less pathogenic PRRSV *in vitro*. The viral life cycle analysis demonstrated that EGCG affected PRRSV replication and assembly, but not viral attachment, entry, or release. Interestingly, EGCG treatment abrogated the increased lipid droplets formation and lipid content induced by PRRSV infection. We further demonstrated that EGCG blocked PRRSV-stimulated expression of the key enzymes in lipid synthesis. In addition, EGCG attenuated PRRSV-induced autophagy that is critical for PRRSV proliferation. The supplementation of oleic acid restored PRRSV replication and assembly under EGCG treatment. Together, our results support that EGCG inhibits PRRSV proliferation through disturbing lipid metabolism.

**IMPORTANCE** Porcine reproductive and respiratory syndrome virus (PRRSV) is an enveloped single-positive-stranded RNA virus that causes acute respiratory distress in piglets and reproductive failure in sows, resulting in huge economic losses to the global swine industry. Several lines of evidence have suggested the crucial roles of lipids in PRRSV proliferation. Our previous report demonstrated that PRRSV activated lipophagy to facilitate viral replication through downregulating the expression of N-Myc downstream-regulated gene 1. The manipulation of lipid metabolism may be a new perspective to prevent PRRSV spread. In the present study, we reported that epigallocatechin-3-gallate (EGCG), the major component of green tea catechins, significantly attenuated PRRSV infection through inhibiting lipid synthesis and autophagy. Given that natural products derived from plants have helped in the prevention and treatment of various infectious diseases, EGCG has a great potential to serve as a safe and environmentally friendly natural compound to treat PRRSV infection.

**KEYWORDS** autophagy, epigallocatechin-3-gallate, lipid synthesis, porcine reproductive and respiratory syndrome virus

Address correspondence to Jiang Wang, wangjiang@henau.edu.cn, Bei-Bei Chu, chubeibei@henau.edu.cn, or Meng-Di Wang, mengdi.OK@163.com.

The authors declare no conflict of interest.

**P**orcine reproductive and respiratory syndrome (PRRS) is a virulent infectious disease of swine and one of the serious infectious diseases in pigs (1, 2). It has led to tremendous economic losses in the swine industry worldwide since its first documentation in the late

1980s (3, 4). PRRS still circulates worldwide, especially in China (5). The etiological agent of PRRS is porcine reproductive and respiratory syndrome virus (PRRSV), an enveloped, positive-sense, single-stranded RNA virus of the family *Arteriviridae* (6, 7). PPRSV especially infects swine and limited cell lines, such as primary *in vivo* target pulmonary alveolar macrophages (PAMs), and the African green monkey kidney epithelial cell line MA-104 and its derivative MARC-145 cells (8, 9). In recent years, the outbreaks of highly virulent variants of PRRSV in China have led to concerns within the global swine industry (10).

Epigallocatechin gallate (EGCG) is the most abundant catechin in green tea, which has a variety of biological functions, including anti-tumor, anti-inflammatory, and immunomodulatory effects (11). It has antiviral activities against various types of viruses, including the human immunodeficiency virus (12), herpes simplex virus (13), and influenza virus (14). Furthermore, EGCG blocks an early step of the hepatitis C virus entry process and inhibits viral cell-to-cell transmission (15). Moreover, it exerts its antiviral effect mainly in the early stage of dengue virus infection by interacting directly with virion (16). Several studies reported that EGCG inhibited PRRSV replication (17–19). The pretreatment of PRRSV with EGCG prevents virus docking to cells (18). EGCG also downregulates the expression of receptors and related proteins required for PRRSV infection (18). However, other possible mechanisms of how EGCG inhibits PRRSV infection have not been well documented.

Lipids are involved in the entire viral life cycle, such as entry, uncoating, genome replication, assembly, and progeny virion release (20). Both cellular and viral lipid rafts are critical for PRRSV infection (21, 22). Free fatty acids (FFAs) are also necessary for optimal PRRSV replication (23). Cholesterol derivative 25-hydroxycholesterol inhibits PRRSV infection both *in vitro* and *in vivo* (24, 25). Lipid droplets are the intracellular reservoir of triglycerides (TG) and are broken down by lipophagy to release glycerol and FFAs (26). We previously reported that PRRSV downregulates the expression of N-Myc downstream-regulated gene 1 to activate lipophagy for optimal viral replication (27). In this study, we examined the roles of EGCG in lipid metabolism and PRRSV proliferation. We demonstrated that EGCG inhibited lipid synthesis and autophagy, which had a destructive effect on PRRSV proliferation.

## RESULTS

**Effects of EGCG on cell viability, cell cycle, and apoptosis.** We first analyzed the effects of EGCG on cell viability, cell cycle, and apoptosis. MARC-145 and porcine alveolar macrophages (PAM) cells were treated with EGCG (0–10 $\mu$M) for 12–72 h. The cell viability evaluated with cell counting kit-8 (CCK-8) assay indicated that EGCG did not exhibit cytotoxicity to either of the cell types (Fig. 1A and B). Next, we treated cells with EGCG (10 $\mu$M) for 0–36 h and found that the cell cycle progression was normal in MARC-145 and PAM cells (Fig. 1C and D). Additionally, we treated MARC-145 and PAM cells with EGCG (0–10 $\mu$M) for 36 h and analyzed apoptosis by Annexin V/PI staining. The flow cytometry analysis indicated that EGCG did not increase the percentage of apoptotic cells (Fig. 1E and G).

**EGCG inhibits PRRSV infection *in vitro*.** To determine whether EGCG could inhibit PRRSV infection, we used a recombinant PRRSV-GFP strain that expressed GFP to monitor viral replication (27). Fluorescent microscopy detected a significant decrease in the GFP signal in response to EGCG treatment, indicating that EGCG inhibited PRRSV-GFP replication (Fig. 2A and B). To further validate the anti-PRRSV effect of EGCG, we analyzed the levels of PRRSV ORF7 (encoding N protein) mRNA and protein. The qRT-PCR analysis indicated that the transcripts of PRRSV *ORF7* in MARC-145 cells were dramatically downregulated by EGCG treatment (Fig. 2C). Meanwhile, the expression of PRRSV-N protein was also affected in EGCG-treated MARC-145 and PAM cells (Fig. 2D and E). Moreover, we analyzed the antiviral effects of EGCG on the highly pathogenic (HP-PRRSV) and less pathogenic (LP-PRRSV) strains of PRRSV. The viral titration assay demonstrated that EGCG induced the decline in the progeny virus production of HP-PRRSV and LP-PRRSV, both in MARC-145 and PAM cells (Fig. 2F and G). These results suggested that EGCG exerted antiviral activity against PRRSV.

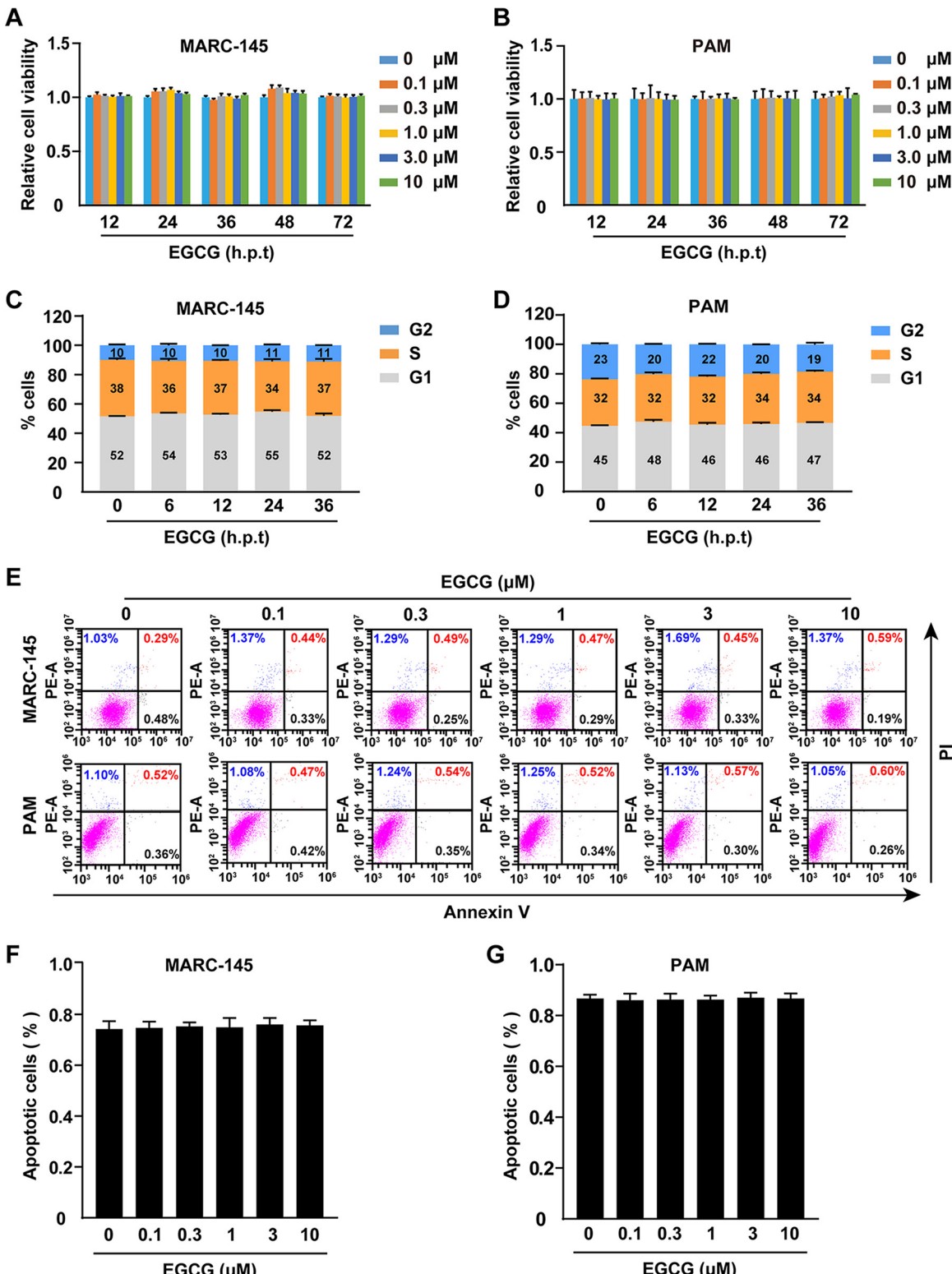

**FIG 1** Effects of EGCG on cell viability, cell cycle, and apoptosis. (A and B) MARC-145 (A) and PAM (B) cells were treated with EGCG (0–10 $\mu$M) for 12–72 h. Cell viability was assessed with CCK-8 assay. h.p.t, hours posttreatment. (C and D) MARC-145 (C) and PAM (D) cells were treated with EGCG (10 $\mu$M) for 0–36 h. The cell cycle was assessed with Hoechst 33342 staining in flow cytometry. (E–G) MARC-145 and PAM cells were treated with EGCG (0–10$\mu$M) for 36 h. Apoptosis was assessed with Annexin V-FITC and PI staining in flow cytometry (E). Quantification of the percentage of cell death was shown in F and G.

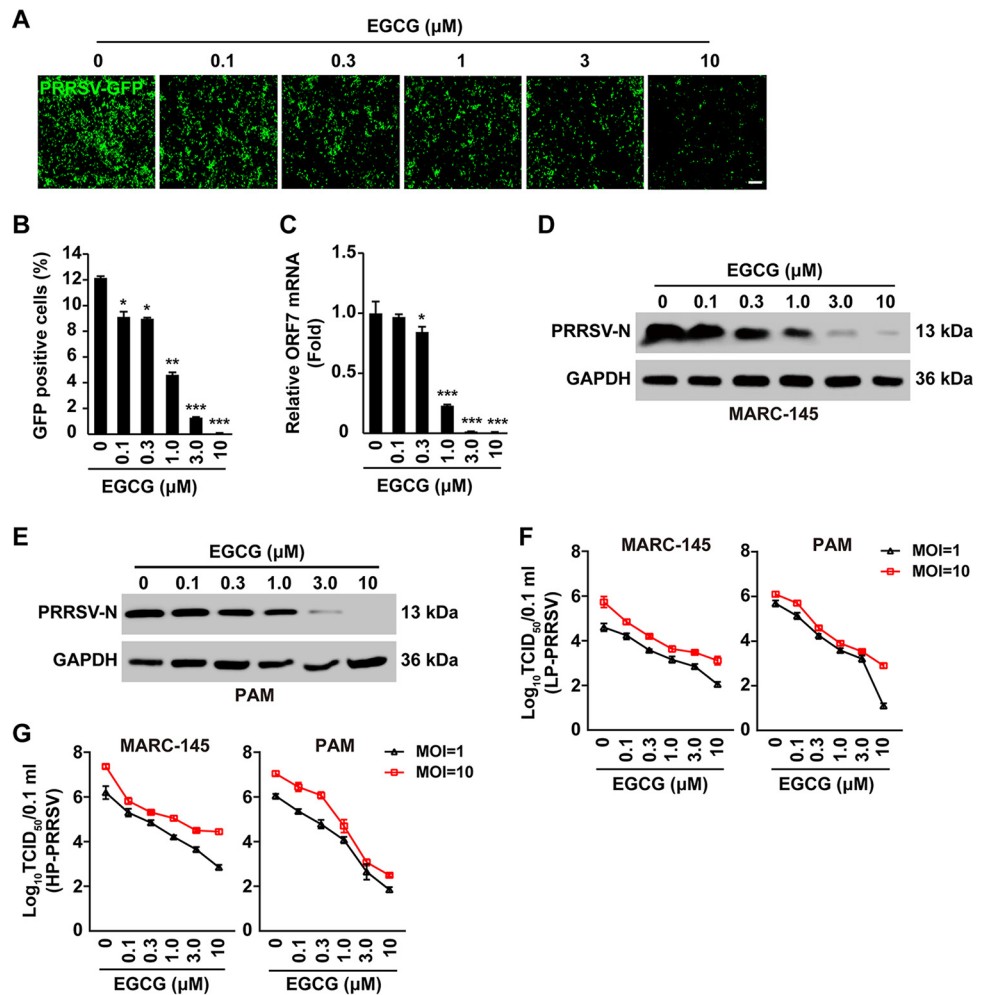

**FIG 2** EGCG inhibited PRRSV proliferation *in vitro*. (A) MARC-145 cells were infected with PRRSV-GFP (MOI = 10) and treated with EGCG (0–10 $\mu$M) for 48 h. Viral replication was analyzed by fluorescence microscopy. Scale bar, 100 $\mu$m. (B) GFP-positive cells from A were analyzed by flow cytometry. *, $P < 0.05$; **, $P < 0.01$; ***, $P < 0.001$. (C) MARC-145 cells were treated as in A. The mRNA level of PRRSV *ORF7* was assessed by qRT-PCR analysis. *, $P < 0.05$; ***, $P < 0.001$. (D and E) MARC-145 (D) and PAM (E) cells were treated as in A. PRRSV-N and GAPDH were assessed by immunoblotting analysis. (F and G) MARC-145 and PAM cells were infected with LP-PRRSV (F, MOI = 10) and HP-PRRSV (G, MOI = 10), and treated with EGCG (0–10 $\mu$M) for 48 h. Viral titers were assessed by TCID$_{50}$ assay.

**EGCG restricts PRRSV replication and assembly.** To dissect the mode of action of EGCG on viral proliferation, we next investigated the impact of EGCG on the life cycle of PRRSV. Viral attachment to host cells is a prerequisite for efficient infection (28). First, HP-PRRSV and EGCG were coincubated on ice for 1 h, and viruses binding to cells were detected by the qRT-PCR analysis of the PRRSV genome with PRRSV *ORF7*-specific primers. As shown in Fig. 3A, equal amounts of *ORF7* mRNA were detected on MARC-145 cells with or without EGCG treatment, suggesting that EGCG did not affect vital attachment. We next sought to determine whether EGCG disturbs viral entry. MARC-145 cells were incubated with HP-PRRSV on ice for 1 h to allow attachment and then were cultured in the absence or presence of EGCG for 2 h at 37°C to allow entry. Invading PRRSV quantified by qRT-PCR analysis of the viral genome demonstrated that EGCG did not influence viral entry (Fig. 3B). In addition, we analyzed PRRSV genome replication by double-stranded RNA (dsRNA) staining and found that EGCG treatment resulted in a significant decrease in dsRNA signaling, indicating that EGCG inhibited PRRSV replication (Fig. 3C and D). We further analyzed viral assembly and release. The viral assembly result indicated that PRRSV particle production was declined in response

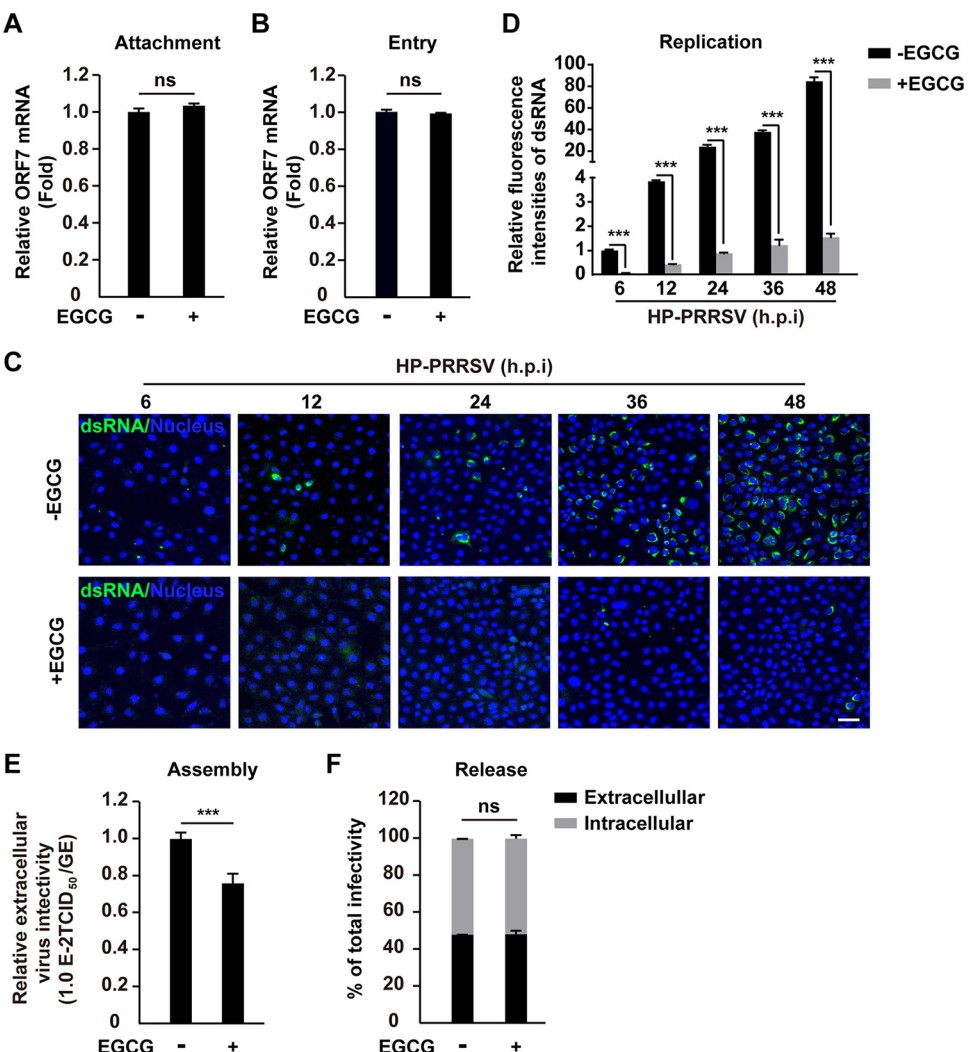

**FIG 3** EGCG impaired PRRSV replication and assembly. (A) HP-PRRSV (MOI = 10) was allowed to bind to the surfaces of MARC-145 cells in the absence or presence of EGCG (10 $\mu$M) on ice for 1 h. After the cells were washed three times with ice-cold PBS, total RNA was extracted and reverse-transcribed to cDNA. Viral attachment was assessed with qRT-PCR analysis of PRRSV *ORF7* mRNA. ns, no significance. (B) HP-PRRSV (MOI = 10) was allowed to bind to the surfaces of MARC-145 cells on ice for 1 h, and the cells were then shifted to 37°C in the absence or presence of EGCG (10 $\mu$M) for 2 h. After the cells were washed three times with PBS, total RNA was extracted and reverse-transcribed to cDNA. Viral entry was assessed with the qRT-PCR analysis of PRRSV *ORF7* mRNA. ns, no significance. (C) MARC-145 cells were infected with HP-PRRSV (MOI = 10) and treated with EGCG (10 $\mu$M) for 6–48 h. Viral replication was assessed with dsRNA staining. Scale bar, 100 $\mu$m. (D) Quantification of the relative fluorescence intensities of dsRNA from C. ***, $P < 0.001$. (E) MARC-145 cells were infected with HP-PRRSV (MOI = 10) and treated with EGCG (10 $\mu$M) for 24 h. The efficiency of viral assembly in the supernatants was determined by comparing the infectious titers (TCID$_{50}$ per milliliter) with the total PRRSV genome equivalents (GE). ***, $P < 0.001$. (F) MARC-145 cells were infected with HP-PRRSV (MOI = 10) and treated with EGCG (10 $\mu$M) for 24 h. The efficiency of virus secretion was determined as the ratio of intra- and extracellular infectivity relative to the total infectivity. ns, no significance.

to EGCG treatment, which was determined by comparing the infectious titers with viral RNA levels (Fig. 3E). However, the infectivity of intracellular and extracellular PRRSV was unaltered by EGCG treatment, suggesting that EGCG had no impact on viral release (Fig. 3F).

**EGCG reduces PRRSV-induced lipid droplet formation and lipid content.** Our previous report indicates that lipid droplets (LDs) are critical for PRRSV replication and assembly (27). Therefore, we examined whether EGCG affected LD formation. MARC-145 and PAM cells were infected with HP-PRRSV and simultaneously treated with EGCG for 0–36 h. Oil Red O and BODIPY staining detected that PRRSV infection increased LD

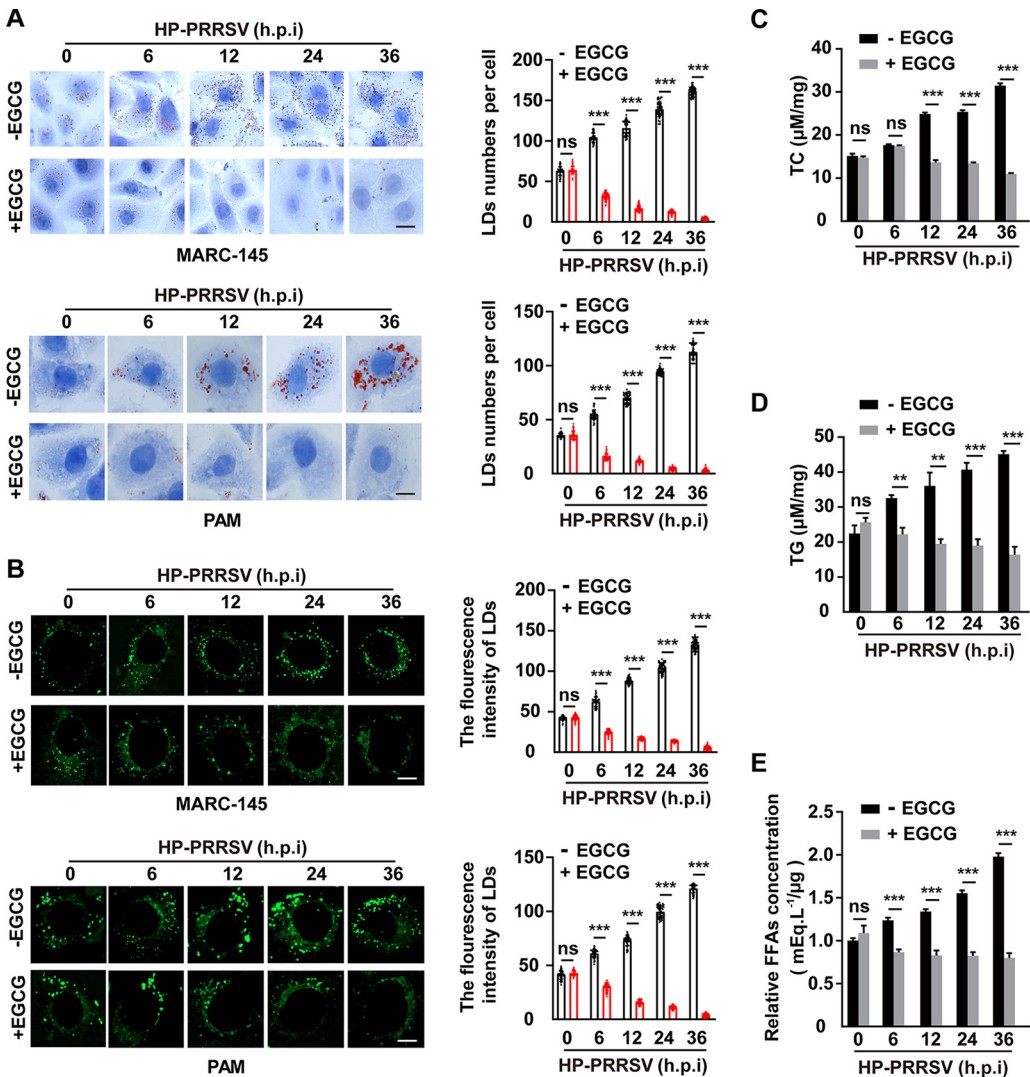

**FIG 4** EGCG reduced lipid droplet formation and lipid content. (A) MARC-145 and PAM cells were infected with HP-PRRSV (MOI = 10) and treated with EGCG (10 μM) for 0–36 h. Lipid droplets were detected by Oil Red O staining. Scale bar, 10 μm. Quantifications of lipid droplets numbers per cell are shown on the right. ***, $P < 0.001$. ns, no significance. (B) MARC-145 and PAM cells were treated as in A. Lipid droplets were detected by BODIPY staining. Scale bar, 10 μm. Quantifications of the fluorescence intensity of lipid droplets are shown on the right. ***, $P < 0.001$. ns, no significance. (C-E) MARC-145 cells were treated as in A. Cellular lipids were extracted, and the amounts of TC (C), TG (D), and FFAs (E) were quantified. **, $P < 0.01$; ***, $P < 0.001$. ns, no significance.

numbers in MARC-145 and PAM cells (Fig. 4A and B). However, EGCG significantly abrogated PRRSV-induced LD formation (Fig. 4A and B). We then measured lipid contents in MARC-145 cells infected with HP-PRRSV and treated with EGCG for 0–36 h. PRRSV enhanced intracellular levels of total cholesterol (TC), TG, and FFAs (Fig. 4C to E). EGCG prevented the generation of these lipids, suggesting that EGCG might inhibit lipid synthesis (Fig. 4C to E). These results demonstrated that EGCG inhibited LD formation and lipid content.

**EGCG inhibits the expression of lipid synthesis-related genes upregulated by PRRSV infection.** We sought to determine whether EGCG could inhibit lipid synthesis. We infected MARC-145 cells with HP-PRRSV and simultaneously treated with EGCG for 0–36 h and then examined the transcription of some key lipogenic genes, such as genes encoding sterol regulatory element-binding transcription factor 1 (*SREBF1*), fatty acid synthase (*FASN*), acetyl-CoA carboxylase alpha (*ACACA*), acetyl-CoA carboxylase beta (*ACACB*), stearoyl-CoA desaturase (*SCD*), sterol regulatory element-binding transcription

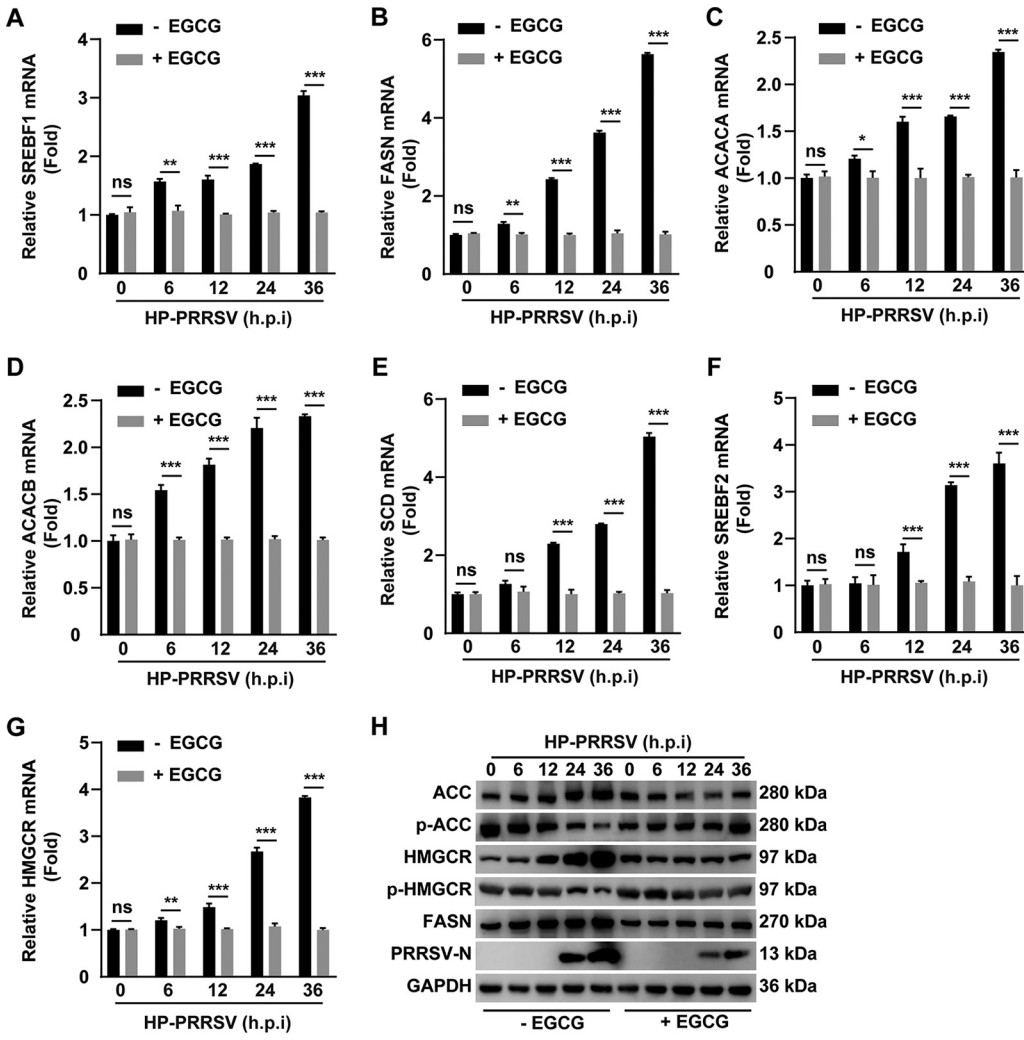

**FIG 5** EGCG inhibited the expression of key factors in lipid synthesis. (A–G) MARC-145 cells were infected with HP-PRRSV (MOI = 10) and treated with EGCG (10 $\mu$M) for 0–36 h. The mRNA levels of *SREBF1* (A), *FASN* (B), *ACACA* (C), *ACACB* (D), *SCD* (E), *SREBF2* (F), and *HMGCR* (G) were assessed by qRT-PCR analysis. *, $P < 0.05$; **, $P < 0.01$; ***, $P < 0.001$. ns, no significance. (H) MARC-145 cells were treated as in A. ACC, phosphorylated ACC; HMGCR, phosphorylated HMGCR. FASN, PRRSV-N, and GAPDH were assessed by immunoblotting analysis.

factor 2 (*SREBF2*), and 3-hydroxy-3-methylglutaryl-CoA reductase (*HMGCR*). The qRT-PCR analysis indicated that PRRSV failed to upregulate the mRNA levels of *SREBF1*, *FASN*, *ACACA*, *ACACB*, *SREBF2*, and *HMGCR* under EGCG treatment (Fig. 5A to G). Furthermore, the immunoblotting analysis indicated that EGCG eliminated the PRRSV-enhanced expression of ACC, HMGCR, FASN, and phosphorylated ACC and HMGCR. (Fig. 5H). These results suggested that EGCG inhibited PRRSV-stimulated lipid synthesis.

**EGCG inhibits PRRSV-induced autophagy.** Autophagy is involved in LD breakdown, which is termed lipophagy (26). Previously, we demonstrated that lipophagy was essential for PRRSV proliferation (27). Therefore, we next sought to determine whether EGCG influenced autophagy. The puncta formation of microtubule-associated protein light chain 3 (LC3) is a hallmark of autophagy induction (29). We observed that LC3 was diffused in the cytosol and nucleus in EGCG treated and untreated cells (Fig. 6A). A dramatic increase in LC3 puncta was detected when the cells were infected with PRRSV (Fig. 6A). EGCG treatment resulted in a significant decline in LC3 puncta in PRRSV-infected cells (Fig. 6A), indicating that EGCG inhibited autophagy. We next analyzed autophagy flux by RFP-GFP-LC3 (30). The colocalization of both GFP and RFP fluorescence represented autophagosomes. The GFP signal was quenched, whereas the

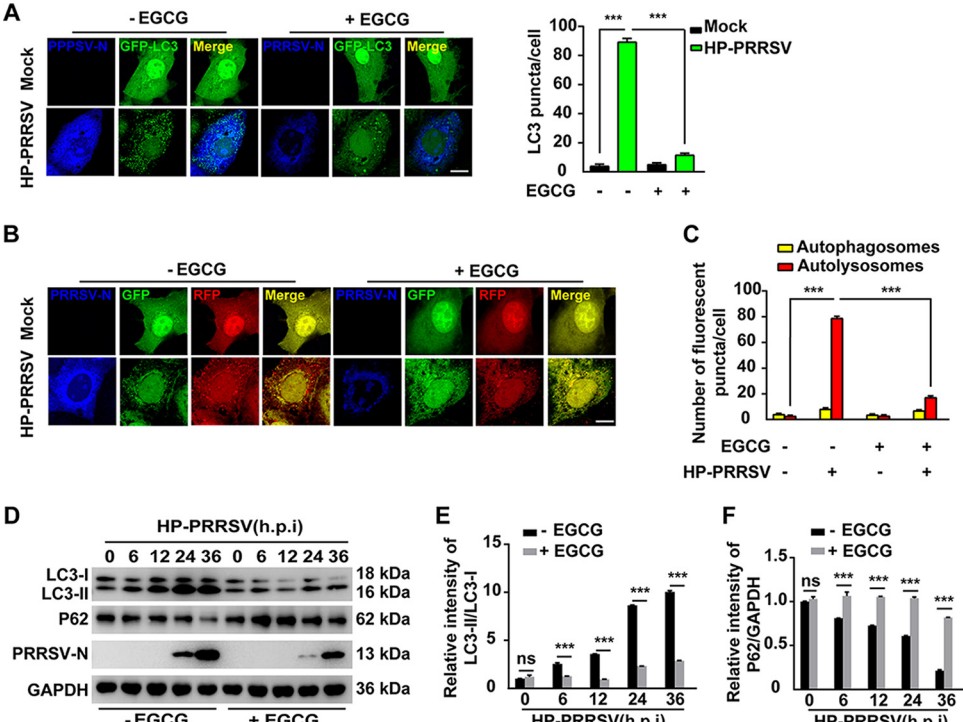

**FIG 6** EGCG inhibited PRRSV-induced autophagy. (A) MARC-145 cells were transfected with plasmid encoding GFP-LC3 for 24 h. The cells were then mock infected or infected with HP-PRRSV (MOI = 10) and treated with EGCG (10 $\mu$M) for 36 h. GFP-LC3 puncta were detected by fluorescence microscopy. Quantification of GFP-LC3 puncta per cell is shown on the right. Scale bar, 10 $\mu$m. ***, $P < 0.001$. (B and C) MARC-145 cells were transfected with plasmid encoding RFP-GFP-LC3 for 24 h. The cells were then mock infected or infected with HP-PRRSV (MOI = 10) and treated with EGCG (10 $\mu$M) for 36 h. The fluorescence of GFP and RFP were detected by fluorescence microscopy (B). Scale bar, 10 $\mu$M. Quantification of autophagosomes (RFP- and GFP-positive puncta) and autolysosomes (only RFP-positive puncta) per cell are shown in C. ***, $P < 0.001$. (D) MARC-145 cells were infected with HP-PRRSV (MOI = 10) and treated with EGCG (10 $\mu$M) for 0–36 h. LC3-I, LC3-II, P62, PRRSV-N, and GAPDH were assessed by immunoblotting analysis. (E) Semi-quantitative densitometric analysis of LC3-II/LC3-I from D was performed with ImageJ software. ***, $P < 0.001$. ns, no significance. (F) Semi-quantitative densitometric analysis of P62/GAPDH from D was performed with ImageJ software. ***, $P < 0.001$. ns, no significance.

RFP signal was stable when autophagosomes fused with lysosomes to form autolysosomes. PRRSV infection enhanced the RFP signal compared with mock infection (Fig. 6B and C), suggesting that PRRSV activated autophagy. When the cells were infected with PRRSV, lower RFP signaling was observed in EGCG-treated cells than that in untreated ones (Fig. 6B and C). These results indicated that EGCG inhibited the autophagy flux. The immunoblotting analysis of LC3-II turnover and P62 degradation is widely used for monitoring autophagy. We found that the relative intensity of LC3-II/LC3-I increased in EGCG-untreated cells, but not in EGCG-treated cells during PRRSV infection (Fig. 6D and E). P62 expression gradually decreased with the prolonged PRRSV infection, but this effect was blocked by EGCG (Fig. 6D and F). All the data indicated that EGCG inhibited autophagy.

**Oleic acid abrogates the inhibitory effect of EGCG on PRRSV proliferation.** We attempted to determine whether the supplementation of lipids with EGCG could rescue PRRSV proliferation. MARC-145 cells were infected with HP-PRRSV and treated with EGCG. Meanwhile, we added oleic acid (OA, 0-150 $\mu$M) to the culture medium as a way for lipid supplementation. We first examined viral replication by dsRNA staining. As shown in Fig. 7A and B, EGCG inhibited PRRSV genome replication, which was rescued by OA supplementation. Notably, 150 $\mu$M OA could fully restore PRRSV replication (Fig. 7A and B). Next, we examined whether OA could rescue PRRSV assembly. In the presence of EGCG, OA treatment resulted in an increase in vial assembly (Fig. 7C). Viral

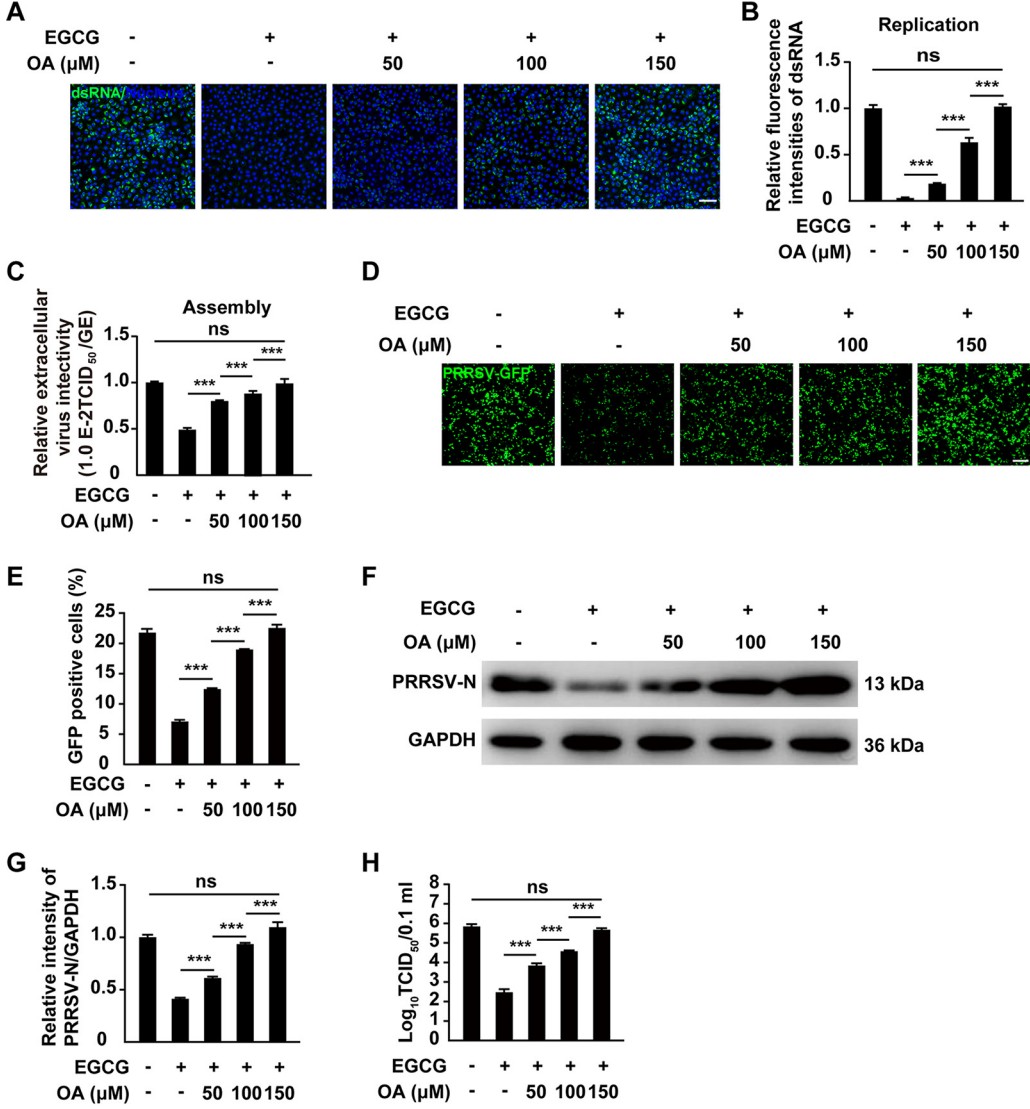

**FIG 7** Oleic acid restored PRRSV replication and assembly in response to EGCG. (A and B) MARC-145 cells were infected with HP-PRRSV (MOI = 10) and treated with EGCG (10 $\mu$M) and OA (0-150 $\mu$M) as indicated for 48 h. Viral replication was assessed with dsRNA staining (A). Quantification of the relative fluorescence intensities of dsRNA is shown in B. ***, $P < 0.001$; ns, no significance. (C) MARC-145 cells were treated as in A. The efficiency of viral assembly in the supernatants was determined by comparing the infectious titers ($TCID_{50}$ per milliliter) with the total PRRSV genome equivalents (GE). ***, $P < 0.001$; ns, no significance. (D and E) MARC-145 cells were infected with PRRSV-GFP (MOI = 10) and treated with EGCG (0–10 $\mu$M) and OA (0–150 $\mu$M) as indicated for 48 h. Viral replication was analyzed by fluorescence microscopy (D). Scale bar, 100 $\mu$m. GFP-positive cells from D were analyzed by flow cytometry (E). ***, $P < 0.001$; ns, no significance. (F) MARC-145 cells were treated as in A. PRRSV-N and GAPDH were assessed by immunoblotting analysis. (G) Semi-quantitative densitometric analysis of PRRSV-N/GAPDH from F was performed with ImageJ software. ***, $P < 0.001$; ns, no significance. (H) MARC-145 cells were treated as in A. Viral titers were assessed by $TCID_{50}$ assay. ***, $P < 0.001$; ns, no significance.

proliferation by monitoring the GFP signal from PRRSV-GFP-infected MARC-145 cells indicated that the viral proliferation was enhanced in an OA dose-dependent manner during EGCG treatment (Fig. 7D and E). The expression of PRRSV-N in cells treated with EGCG and OA (150 $\mu$M) was comparable to that in nontreated cells (Fig. 7F and G). Finally, we determined whether OA could rescue the EGCG-inhibited multiplication of the PRRSV progeny virus using $TCID_{50}$ assay. The viral titration result indicated that the inhibitory effect of EGCG on the production of the PRRSV progeny virus was abrogated by OA (Fig. 7H). These results further suggested that EGCG restricted PRRSV proliferation by inhibiting lipid synthesis.

## DISCUSSION

Lipid plays important roles in PRRSV proliferation. Lipid rafts are plasma membrane microdomains enriched in cholesterol and sphingolipids involved in the lateral compartmentalization of molecules on the cell surface (31). Several lines of evidence have suggested that lipid rafts are required for PRRSV entry (21, 22, 32). FFAs play positive roles in PRRSV proliferation (23), but 25-hydroxycholesterol impairs PRRSV infection (24, 25). However, how lipids participate in the entire life cycle of PRRSV is still not well understood. Previously, we indicated that PRRSV drove lipolysis by lipophagy to facilitate optimal viral replication (27). In the present study, we demonstrated that PRRSV infection enhanced lipid synthesis, which was prevented by EGCG. Lipid may be the supplier of either energy requirements or building blocks for PRRSV replication and assembly. Therefore, fine tuning lipid synthesis is a promising therapeutic to prevent and control PRRSV spread.

Autophagy is a self-degradative process to maintain cellular homeostasis (33). It is believed to have evolved to combat infection by a number of intracellular pathogens (34). Nevertheless, autophagy induction benefits PRRSV proliferation (35–37). PRRSV NSP3, NSP5, and NSP9 play pivotal roles in activating autophagy (38). PRRSV infection causes endoplasmic reticulum (ER) stress to trigger the unfolded protein response responsible for the induction of autophagy (39–41). In addition, PRRSV can induce autophagy through the PI3K/Akt/mTOR signaling pathway (42–44). It can also trigger mitochondrial fission and mitophagy to attenuate apoptosis (45). We suggested that PRRSV activated lipophagy through downregulating N-Myc downstream-regulated gene 1 (27). Our present data indicated that EGCG suppressed PRRSV-induced autophagy. However, Meng and colleagues demonstrate that EGCG induced autophagy by targeting the mTOR pathway in vascular endothelial cells (46). EGCG induces autophagy and promotes cell survival via shifting the balance of mTOR-AMPK pathways in ER stress (47). The mechanism of how EGCG inhibits autophagy during PRRSV infection needs further investigation.

PRRS was first recognized in the United States in 1987 and is the most economically important infectious disease of pigs worldwide (1). Conventional vaccines of PRRSV do not provide satisfactory or sustainable prevention. The development of effective antivirals against PRRSV is a new strategy to prevent PRRS spread. Catechins are the main components of tea polyphenols (48). They exert antiviral activities against PRRSV through suppressing viral attachment, internalization, and limiting the synthesis of viral nonstructural protein 2 (49). Here, we reported a distinct mechanism by which EGCG inhibited PRRSV-induced lipid synthesis and autophagy, which were critical for viral replication and assembly. The treatment of cells with OA restored the anti-PRRSV effect of EGCG, indicating that EGCG inhibited PRRSV proliferation by disturbing lipid metabolism.

## MATERIALS AND METHODS

**Reagents.** SYBR Premix *Ex Taq* (RR420A) and TRIzol Reagent (D9108B) were ordered from TaKaRa (Otsu, Shiga, Japan); a TIANamp Virus RNA/DNA Kit (DP315-R) was ordered from Tiangen (Beijing, China); a Dead Cell Apoptosis Kit with Annexin V-fluorescein (FITC) and PI (V13242), and BODIPY 493/503 (D3922) were ordered from Thermo Fisher Scientific (MA, USA); Hoechst 33342 (561908) was ordered from BD Biosciences (NJ, USA); Oil Red O (O0625) was ordered from Sigma-Aldrich (MO, USA); EGCG (HY-13653) was ordered from MedChemExpress (NJ, USA); a LabAssay nonessential fatty acid (294-63601) assay kit for FFAs was ordered from Wako Bioproducts (VA, USA); and a triglyceride assay kit (E1013) and a cholesterol assay kit (E1015) were ordered from Applygen Technologies Inc. (Beijing, China).

**Antibodies.** The antibodies anti-LC3 (#4599), anti-P62 (#5114), anti-ACC (#3676), and antiphosphorylated ACC (#11818) were ordered from Cell Signaling Technology (MA, USA); anti-GAPDH (10494-1-AP) was ordered from Proteintech (Wuhan, China); anti-FASN (ab22759) was ordered from Abcam (MA, USA); anti-HMGCR (MABS1233) was ordered from Sigma-Aldrich; anti-phosphorylated HMGCR (bs-4063R) was ordered from Bioss (MA, USA); anti-PRRSV nucleocapsid (N) SDOW17 was ordered from Rural Technologies (WA, USA); anti-dsRNA (J2 and K1) was ordered from English & Scientific Consulting (Szirak, Hungary); horseradish peroxidase (HRP)-conjugated donkey anti-mouse IgG (715-035-150) and anti-rabbit IgG (711-035-152) were ordered from Jackson ImmunoResearch Laboratories (West Grove, PA, USA); and anti-mouse IgG labeled with Alexa Fluor 555 (A21424), Alexa Fluor 488 (A21429), and anti-rabbit IgG labeled with Alexa Fluor 488 (A11034) were ordered from Thermo Fisher Scientific. The aforementioned antibodies were used at dilutions of 1:500 for immunofluorescence staining and 1:1,000 for immunoblotting analysis.

**Cells and viruses.** MARC-145 cells were cultured in Dulbecco's modified Eagle's medium (DMEM, 10566-016, Gibco, NY, USA) supplemented with 10% fetal bovine serum (FBS, 10099141C, Gibco), 100

units/mL penicillin, and 100 $\mu$g/mL streptomycin sulfate (B540732, Sangon Biotech, Shanghai, China). PAMs were obtained and cultured as previously described (27). The recombinant PRRSV-GFP strain (50) was kindly donated by Professor En-Min Zhou from Northwest A&F University (Yangling, Shaanxi, China). The highly pathogenic PRRSV strain HN07-1 (HP-PRRSV, GenBank accession no. KX766378.1) (51) was kindly donated by Professor Gai-Ping Zhang from Henan Agricultural University (Zhengzhou, China). The less pathogenic PRRSV strain BJ-4 (LP-PRRSV no. AF331831.1) was used as described previously (27). Viruses were propagated and titrated in MARC-145 cells by calculating the median tissue culture infective dose as previously described (27).

**Plasmids and transient transfection.** The plasmids pEGFP-LC3 (#24920) and pMRX-IP-GFP-LC3-RFP (#84573) were purchased from Addgene (MA, USA) and used as previously described (27). All plasmids were transfected with Lipofectamine 3000 (L3000015, Invitrogen, NY, USA) following the manufacturer's protocols.

**Cell viability analysis.** The cell viability was evaluated with CCK-8 assay (GK3607, DingGuo, Beijing, China). The cells were seeded at $1 \times 10^4$ per well into 96-well plates. On the next day, the medium was changed to DMEM/10% FBS supplemented with EGCG (0–10 $\mu$M), and the cells were cultured for 12–72 h. CCK-8 (10 $\mu$L) was then added to each well, and the cells were incubated for 3 h at 37°C. The absorbance was detected at 450 nm with a microplate reader (Varioskan Flash, Thermo Fisher Scientific).

**Cell cycle analysis.** The cells were seeded at $1.2 \times 10^5$ per well into 24-well plates. On the following day, the medium was changed to DMEM/10% FBS supplemented with DMSO and EGCG (10 $\mu$M) for the indicated times. The cells were digested with trypsin-EDTA (25200072, Gibco) and resuspended in phosphate-buffered saline (PBS) containing 5 $\mu$g/mL of Hoechst 33342. After incubation for 1 h at 37°C, the cell cycle profiles were collected by flow cytometry on a CytoFLEX instrument (Beckman Coulter, CA, USA). Data were analyzed with FlowJo software.

**Apoptosis analysis.** The cells were seeded at $1.2 \times 10^5$ per well into 24-well plates. On the following day, the medium was changed to DMEM/10% FBS supplemented with DMSO and EGCG (0–10 $\mu$M) for 36 h. Annexin V/PI staining was performed with the Dead Cell Apoptosis Kit with Annexin V-FITC and PI (Thermo Fisher Scientific). The percentage of dead cells (positive for both Annexin V and PI) was measured by flow cytometry on a CytoFLEX instrument (Beckman Coulter). Data were analyzed with FlowJo software.

**qRT-PCR analysis.** Total RNA was extracted from cells using the TRIzol Reagent (9108, TaKaRa) and reverse-transcribed to cDNA using a PrimeScript RT reagent kit (RR047A, TaKaRa) following the manufacturers' protocols. qRT-PCR was performed in triplicate using SYBR Premix *Ex Taq* (RR820A, TaKaRa) following the manufacturer's protocols. The results were normalized against the level of $\beta$-actin expression. The amounts were quantified by the $2^{-\Delta\Delta Ct}$ method. The primers used for qRT-PCR were as follows: *$\beta$-actin*-Fw: 5′-CGTGGACATCCGTAAAGAC-3′; *$\beta$-actin*-Rv: 5′-GGAAGGTGGACAGCGAGGC-3′; *FASN*-Fw: 5′-GCACACTT ACGTACTGGCCT-3′; *FASN*-Rv: 5′-TGATGATTAGGTCCACGGCG-3′; *SCD*-Fw: 5′-GCCGGAGTTTACAGAAGCCT-3′; *SCD*-Rv: 5′-GGTGACCGTGTCCGGTATTT-3′; *ACACA*-Fw: 5′-CCGCTTGCCTGTCTTTTGAT-3′; *ACACA*-Rv: 5′-AC GTTATCCCCAAACCCAGG-3′; *ACACB*-Fw: 5′-GCTCTAGGACCGCATCTCG-3′; *ACACB*-Rv: 5′-CATGCTCGGCTT TACTTCGC-3′; *HMGCR*-Fw: 5′-ATTGTGTGCGGGACCGTAAT-3′; *HMGCR*-Rv: 5′-AATGCCCGTGTTCCAGTTCA-3′; *SREBF1*-Fw: 5′-GACGAGCCACCCTTCAGCAA-3′; *SREBF1*-Rv: 5′-GCATGTCTTCGAACGTGCAAT-3′; *SREBF2*-Fw: 5′-TTGTCGGGTGTCATGGGCG-3′; *SREBF2*-Rv: 5′-ATTGCAGCATCTCGTCGATGT-3′.

**Immunoblotting analysis.** The cells were lysed in lysis buffer (50 mM Tris-HCl, pH 8.0, 150 mM NaCl, 1% Triton X-100, 1% sodium deoxycholate, 0.1% SDS, and 2 mM $MgCl_2$) supplemented with a protease and phosphatase inhibitor cocktail (HY-K0010 and HY-K0022, MedChemExpress). Protein samples were separated by SDS-PAGE and then transferred to polypropylene fluoride membranes (C3117, Millipore, MA, USA). After incubation in 5% nonfat milk (A600669, Sangon Biotech) for 30 min, the membrane was incubated with the primary antibody overnight at 4°C and then with the appropriate HRP-conjugated secondary antibody for 1 h at room temperature. The target proteins were detected with the Luminata Crescendo immunoblotting HRP substrate (WBLUR0500, Millipore) on a GE AI600 imaging system.

**Immunofluorescence analysis.** The cells grown on coverslips were fixed with 4% paraformaldehyde in PBS for 30 min at room temperature. The cells were permeabilized in PBS/0.1% Triton X-100 and blocked with PBS/10% FBS for 1 h at room temperature. After the cells were washed three times with PBS, the primary antibodies were incubated with the cells for 1 h at room temperature. The cells were then incubated with the appropriate Alexa-Fluor-conjugated secondary antibodies for 1 h at room temperature. The images were captured on a Zeiss (Oberkochen, Germany) LSM 800 confocal microscope.

**Viral assembly assay.** MARC-145 cells were infected with HP-PRRSV (MOI = 10). The efficiency of viral assembly in the supernatants was determined by comparing the infectious titers ($TCID_{50}$/mL) with the total PRRSV genome equivalents (GE).

**Determination of intracellular FFAs.** FFAs were measured with the LabAssay nonessential fatty acid kit (Wako Bioproducts), following the manufacturer's protocols. The values were normalized to the total cellular protein content.

**Determination of intracellular triglyceride and cholesterol.** Intracellular triglycerides and cholesterol were measured with the triglyceride assay kit (Applygen Technologies Inc.) and the cholesterol assay kit (Applygen Technologies Inc.), following the manufacturer's protocols. The values were normalized to the total cellular protein content.

**Oil Red O staining.** The cells were fixed in 4% paraformaldehyde for 30 min and then incubated in Oil Red O (3 $\mu$g/mL) for 15 min. They were washed with 70% alcohol for 5 s to remove any background stain, rinsed in double-distilled Millipore water, counterstained with Harris hematoxylin, washed, mounted, and observed under a light microscope. The LD number was determined with the ImageJ "analyze particles" function (areas of particles < 0.01 mm$^2$ were excluded).

**BODIPY staining.** The cells were fixed in 4% PFA for 30 min and then incubated with 2 $\mu$g/mL BODIPY 493/503 (493 nm excitation/503 nm emission) for 30 min. The digital images were obtained with a Zeiss LSM800 confocal microscope. The fluorescence intensity was determined with ImageJ software.

**Statistical analysis.** Data were obtained from at least three independent experiments for quantitative analyses and expressed as means $\pm$ standard errors of the means. All statistical analyses were performed with the two-tailed Student's $t$ test. Significant differences relative to the corresponding controls were accepted at *, $P < 0.05$; **, $P < 0.01$; and ***, $P < 0.001$.

## ACKNOWLEDGMENTS

This study was supported by grants from the National Natural Science Foundation of China (32072858, 31902284), the Youth Backbone Teachers Training Plan for Universities in Henan Province (2020GGJS258), and the Doctoral Research Initiation Fund of Henan University of Animal Husbandry and Economy (2019HNUAHEDF040).

We declare no conflicts of interest.

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
