## [Reviewer comments · Microbiology Spectrum]

Microbiology Spectrum

EGCG restricts PRRSV proliferation by disturbing lipid metabolism

Meng-Di Wang, Peng-Wei Yu, Peng-fei Fu, Lei Zeng, Yan-Li Qi, Xiu-Qing Li, Qi Wang, Guo-Yu Yang, Huawei Li, Jiang Wang, and Bei-Bei Chu

Corresponding Author(s): Meng-Di Wang, Henan University of Animal Husbandry and Economy

Review Timeline:

Submission Date:	November 15, 2021
Editorial Decision:	January 27, 2022
Revision Received:	March 3, 2022
Accepted:	March 20, 2022

Editor: Laura Delgui

Reviewer(s): The reviewers have opted to remain anonymous.

Transaction Report:

DOI: <https://doi.org/10.1128/spectrum.02276-21>

January 27, 2022

Dr. Meng Di Wang
Henan University of Animal Husbandry and Economy
Longzi Hubei Road, Zhengzhou city
zhengzhou
China

Re: Spectrum02276-21 (EGCG restricts PRRSV proliferation by disturbing lipid metabolism)

Dear Dr. Meng Di Wang:

Dear author, thanks for submitting your manuscript to Microbiology Spectrum. Based on the reviewer #1 comments, please expand your methodology and conclusions. If those items are properly modified, we would be happy to re-consider publishing your manuscript in Microbiology Spectrum.

Link Not Available

Sincerely,

Laura Delgui

Journals Department
Reviewer comments:

Reviewer #1 (Comments for the Author):

In this manuscript, Yu and colleagues investigated the anti-PRRSV role of EGCG, the most abundant catechin in green tea, and found that EGCG exhibits potent anti-PRRSV activity through attenuating lipid synthesis and PRRSV-induced autophagy. The antiviral effects of EGCG have been extensively reported against many diverse viruses. Although the findings in this manuscript are novel, some conclusions are inconsistent with previous studies. Additional experiments should be performed to support their conclusions.

(1) In a previous study (ref 18), the anti-PRRSV role of EGCG has been demonstrated, however, no significant anti-PRRSV activity could be observed when the concentration of EGCG is less than 31.25 μM . In this manuscript, 10 μM of EGCG reduced the viral titer by nearly 4 logs. Please explain these different results.

(2) Previous studies have showed that EGCG-treated PRRSV particles are impaired in binding to target cells, indicating that

EGCG should affect the attachment step of PRRSV infection; Also, EGCG treatment downregulates the expression of CD163, the key receptor for PRRSV entry, indicating that EGCG should affect the entry step of PRRSV infection. However, in this manuscript, the authors found that EGCG inhibits the replication and assembly of PRRSV, while not the attachment, entry and release steps. Indeed, the studies of EGCG effects on different viruses showed that EGCG not only acts directly on the virion surface proteins (eg. Influenza virus HA), as well as the viral proteases (eg. SARS-CoV-2 3C-like protease), but also indirectly blocks viral attachment by competing with heparan sulfate or sialic acid moieties in cellular glycans for virion binding. It is well-known that PRRSV utilizes heparan sulfate or sialic acid for viral attachment. Please explain why EGCG has no influence on the attachment and entry processes of PRRSV infection.

(3) The authors showed that EGCG attenuates PRRSV-induced autophagy, which is a critical result to support their conclusion that EGCG restricts PRRSV proliferation by disturbing lipid metabolism. However, many studies have demonstrated that EGCG can induce autophagy in different cells. Whether the phenomenon that EGCG induces autophagy is limited to a particular cell type (such as MARC-145 cells in this manuscript)?

(4) In Figure 2B, the infectivity of PRRSV is only 12% at 48 hpi (no-EGCG treatment group). This infection ratio is very low. Under the same infection condition, the authors detected the dsRNA and showed that the relative fluorescence intensities of dsRNA are up to 80% (Figure 3C and 3D). Please explain these different results.

(5) The authors only analyzed the mRNA expressions of some key lipogenic genes and concluded that EGCG inhibited the expression of lipid synthesis-related genes upregulated by PRRSV infection. The protein levels of lipid synthesis-related genes should be detected through western blot assays. In addition, the phosphorylation levels of ACACA and HMGCR should be analyzed.

(6) Please provide a more detailed protocol for detecting viral assembly in the Materials and Methods.

(7) Line 122, "MAR-145" should be changed to "MARC-145"; Line 175, "lypophagy" should be changed to "lipophagy"; Line 285, "5'" should be changed to "5'-".

Reviewer #2 (Comments for the Author):

In the paper, the authors found that EGCG attenuated PRRSV-induced autophagy for PRRSV proliferation. Under EGCG treatment, the supplementation of oleic acid restored PRRSV replication and assembly. The finding of the paper is interesting and provides some new insights for mechanism of PRRSV replication. Moreover, it also gives a new strategy for design of anti-PRRSV drugs. However, the writing of the manuscript should be revised. For example, the "-" should be same throughout the manuscript. Some initiative sentences for characterization of materials should be changed to passive ones in the manuscript. In the discussion section, the conclusion should be added in the end.

Staff Comments:

Preparing Revision Guidelines

Please return the manuscript within 60 days; if you cannot complete the modification within this time period, please contact me. If you do not wish to modify the manuscript and prefer to submit it to another journal, please notify me of your decision immediately so that the manuscript may be formally withdrawn from consideration by Microbiology Spectrum.

Reviewer #1 (Comments for the Author):

In this manuscript, Yu and colleagues investigated the anti-PRRSV role of EGCG, the most abundant catechin in green tea, and found that EGCG exhibits potent anti-PRRSV activity through attenuating lipid synthesis and PRRSV-induced autophagy. The antiviral effects of EGCG have been extensively reported against many diverse viruses. Although the findings in this manuscript are novel, some conclusions are inconsistent with previous studies. Additional experiments should be performed to support their conclusions.

(1) In a previous study (ref 18), the anti-PRRSV role of EGCG has been demonstrated, however, no significant anti-PRRSV activity could be observed when the concentration of EGCG is less than 31.25 μ M. In this manuscript, 10 μ M of EGCG reduced the viral titer by nearly 4 logs. Please explain these different results.

Response: We examined the anti-PRRSV effect of EGCG by recombinant PRRSV-GFP, qRT-PCR of PRRSV ORF7 mRNA, immunoblotting of PRRSV N and TCID₅₀ assay of both LP-PRRSV and HP-PRRSV. All the results demonstrated that 10 μ M of EGCG could inhibit PRRSV proliferation. The differences between our and previous results may be due to two possibilities. First, PRRSV strain JXA1-R is used in previous study [1], while PRRSV strains HN07-1 and BJ-4 were used in our present study. Different PRRSV strains may exhibit diverse sensitivity to EGCG. Second, EGCG was purchased from Santa Cruz (sc-200802, purity \geq 98%) in previous study [1], while EGCG was purchased from MedChemExpress (HY-13653, purity 99.87%) in our present study. The different EGCG sources may exhibit differential anti-PRRSV effect due to the purity.

(2) Previous studies have showed that EGCG-treated PRRSV particles are impaired in binding to target cells, indicating that EGCG should affect the attachment step of PRRSV infection; Also, EGCG treatment downregulates the expression of CD163, the key receptor for PRRSV entry, indicating that EGCG should affect the entry step of PRRSV infection. However, in this manuscript, the authors found that EGCG inhibits the replication and assembly of PRRSV, while not the attachment, entry and release

steps. Indeed, the studies of EGCG effects on different viruses showed that EGCG not only acts directly on the virion surface proteins (eg. Influenza virus HA), as well as the viral proteases (eg. SARS-CoV-2 3C-like protease), but also indirectly blocks viral attachment by competing with heparan sulfate or sialic acid moieties in cellular glycans for virion binding. It is well-known that PRRSV utilizes heparan sulfate or sialic acid for viral attachment. Please explain why EGCG has no influence on the attachment and entry processes of PRRSV infection.

Response: Cells are pre-treated with EGCG for 6-24 h and then infected with PRRSV in previous study [1]. In our viral attachment assay, HP-PRRSV (MOI = 10) was allowed to bind to the surfaces of MARC-145 cells in the absence or presence of EGCG (10 μ M) on ice for 1 h. After the cells were washed three times with ice-cold PBS, total RNA was extracted and reverse-transcribed to cDNA. Viral attachment was assessed with qRT-PCR analysis of PRRSV ORF7 mRNA. Attachment assay usually performed at 4°C or on ice, because the endocytic machineries are inhibited at low temperature. In addition, we did not pre-treat cells with EGCG when we perform viral attachment assay. EGCG treatment on ice for 1 h is not enough long to modulate CD163 expression, thereby inhibiting PRRSV attachment.

The reason why EGCG has no influence on the entry processes of PRRSV infection may be due to the different procedures to treat cells. Colpitts and colleagues have indicated EGCG inhibits virion attachment to heparan Sulfate- or sialic acid-containing glycans [2]. In their binding assay, R18-labeled HSV-1, VSV, HCV, IAV, or VACV virions ($\sim 1 \times 10^4$ PFU or FFU) were exposed for 10 min at 37°C to EGCG, DMSO, vehicle or 100 μ g/ml heparin in phenol red-free DMEM (pH 7.2). In our binding assay, we did not pre-treat cells with EGCG when we perform viral entry assay. In addition, Colpitts and colleagues treat virus and cells with log 0-4 μ M of EGCG. The maximal concentration of EGCG we used to treat cells was 10 μ M, which was might be not efficient to compete with heparan sulfate or sialic acid moieties in cellular glycans for viral attachment and entry.

(3) The authors showed that EGCG attenuates PRRSV-induced autophagy, which is a

critical result to support their conclusion that EGCG restricts PRRSV proliferation by disturbing lipid metabolism. However, many studies have demonstrated that EGCG can induce autophagy in different cells. Whether the phenomenon that EGCG induces autophagy is limited to a particular cell type (such as MARC-145 cells in this manuscript)?

Response: Our previous study has demonstrated that PRRSV stimulates lipophagy to facilitate its replication and assembly [3], suggesting lipids serve as building block and energy source for PRRSV infection. Autophagosome biogenesis involves de novo formation of a membrane that elongates to sequester cytoplasmic cargo and closes to form a double-membrane vesicle [4]. Mammalian autophagosomes can originate concomitantly at several sites that are closely associated with specific phosphatidylinositol 3-phosphate-enriched subdomains of the ER, referred to as omegasomes [5]. Further elongation of the phagophore membrane seems to involve several membrane sources. Lipids are the major component of membrane, so they are pivotal for autophagosome biogenesis.

In the present study, we indicated that EGCG attenuates PRRSV-induced autophagy. We postulated that EGCG inhibited lipid synthesis and PRRSV replication consumed a large amount of cellular lipids, both of which dramatically decreased cellular lipid source that was required for autophagosome biogenesis and subsequent autophagy induction. Whether the phenomenon that EGCG induces autophagy is limited to MARC-145 cells needs for future investigation.

(4) In Figure 2B, the infectivity of PRRSV is only 12% at 48 hpi (no-EGCG treatment group). This infection ratio is very low. Under the same infection condition, the authors detected the dsRNA and showed that the relative fluorescence intensities of dsRNA are up to 80% (Figure 3C and 3D). Please explain these different results.

Response: In Figure 2B, MARC-145 cells were infected with PRRSV-GFP (MOI = 10) and treated with EGCG (0–10 μ M) for 48 h. The percentage of GFP-positive cells (represented viral proliferation) in all cells was analyzed by flow cytometry. In Figure 3C and 3D, MARC-145 cells were infected with HP-PRRSV (MOI = 10) and treated

with EGCG (10 μ M) for 6-48 h. Viral replication was assessed with dsRNA staining. We set the fluorescence intensity of dsRNA in the absence of EGCG at 6 h post PRRSV infection as 1 and the relative fluorescence intensities were quantified in other treatment. These two quantification methods are used in our previous publication [3].

Figure 3G [3]

Figure 5C and 5D [3]

(5) *The authors only analyzed the mRNA expressions of some key lipogenic genes and concluded that EGCG inhibited the expression of lipid synthesis-related genes upregulated by PRRSV infection. The protein levels of lipid synthesis-related genes should be detected through western blot assays. In addition, the phosphorylation levels of ACACA and HMGCR should be analyzed.*

Response: We have analyzed the phosphorylation levels of ACACA and HMGCR according to the reviewer's suggestion.

(6) *Please provide a more detailed protocol for detecting viral assembly in the Materials and Methods.*

Response: We thank for the reviewer's suggestion. We have provided a more detailed

protocol for detecting viral assembly in the Materials and Methods as follow: MARC-145 cells were infected with HP-PRRSV (MOI = 10). The efficiency of viral assembly in the supernatants was determined by comparing the infectious titers (TCID₅₀/ml) with the total PRRSV genome equivalents (GE) [6].

(7) Line 122, "MAR-145" should be changed to "MARC-145"; Line 175, "lypophagy" should be changed to "lipophagy"; Line 285, "5'" should be changed to "5'-'".

Response: We thank for the reviewer's comments. We have made the changes according to the reviewer's suggestion.

References:

1. Ge, M.; Xiao, Y.; Chen, H.; Luo, F.; Du, G.; Zeng, F., Multiple antiviral approaches of (-)-epigallocatechin-3-gallate (EGCG) against porcine reproductive and respiratory syndrome virus infection in vitro. *Antiviral Res* **2018**, 158, 52-62.
2. Colpitts, C. C.; Schang, L. M., A small molecule inhibits virion attachment to heparan sulfate- or sialic acid-containing glycans. *J Virol* **2014**, 88, (14), 7806-17.
3. Wang, J.; Liu, J. Y.; Shao, K. Y.; Han, Y. Q.; Li, G. L.; Ming, S. L.; Su, B. Q.; Du, Y. K.; Liu, Z. H.; Zhang, G. P.; Yang, G. Y.; Chu, B. B., Porcine Reproductive and Respiratory Syndrome Virus Activates Lipophagy To Facilitate Viral Replication through Downregulation of NDRG1 Expression. *J Virol* **2019**, 93, (17).
4. Melia, T. J.; Lystad, A. H.; Simonsen, A., Autophagosome biogenesis: From membrane growth to closure. *J Cell Biol* **2020**, 219, (6).
5. Axe, E. L.; Walker, S. A.; Manifava, M.; Chandra, P.; Roderick, H. L.; Habermann, A.; Griffiths, G.; Ktistakis, N. T., Autophagosome formation from membrane compartments enriched in phosphatidylinositol 3-phosphate and dynamically connected to the endoplasmic reticulum. *J Cell Biol* **2008**, 182, (4), 685-701.
6. Cai, H.; Yao, W.; Li, L.; Li, X.; Hu, L.; Mai, R.; Peng, T., Cell-death-inducing DFFA-like Effector B Contributes to the Assembly of Hepatitis C Virus (HCV) Particles and Interacts with HCV NS5A. *Sci Rep* **2016**, 6, 27778.

Reviewer #2 (Comments for the Author):

In the paper, the authors found that EGCG attenuated PRRSV-induced autophagy for PRRSV proliferation. Under EGCG treatment, the supplementation of oleic acid restored PRRSV replication and assembly. The finding of the paper is interesting and provides some new insights for mechanism of PRRSV replication. Moreover, it also gives a new strategy for design of anti-PRRSV drugs. However, the writing of the manuscript should be revised. For example, the "-" should be same throughout the manuscript. Some initiative sentences for characterization of materials should be changed to passive ones in the manuscript. In the discussion section, the conclusion should be added in the end.

Response: We thank for the reviewer's comments. We have made the changes according to the reviewer's suggestion.

March 20, 2022

Dr. Meng-Di Wang
Henan University of Animal Husbandry and Economy
zhengzhou
China

Re: Spectrum02276-21R1 (EGCG restricts PRRSV proliferation by disturbing lipid metabolism)

Dear Dr. Meng-Di Wang:

Manuscript Spectrum02276-21R1 has been revised by the reviewer who had previously made some critical observations. In this second round of revision, he considered that the authors have addressed his concerns adequately.

Your manuscript has been accepted, and I am forwarding it to the ASM Journals Department for publication. You will be notified when your proofs are ready to be viewed.

Sincerely,

Laura Delgui
Editor, Microbiology Spectrum
